# Human Milk Fatty Acid Composition and Its Effect on Preterm Infants’ Growth Velocity

**DOI:** 10.3390/children10060939

**Published:** 2023-05-26

**Authors:** Batool Ahmed, Afnan Freije, Amina Omran, Mariangela Rondanelli, Mirko Marino, Simone Perna

**Affiliations:** 1Department of Biology, College of Science, University of Bahrain, Sakhir 32038, Bahrain; 2IRCCS Mondino Foundation, 27100 Pavia, Italy; 3Unit of Human and Clinical Nutrition, Department of Public Health, Experimental and Forensic Medicine, University of Pavia, 27100 Pavia, Italy; 4Department of Food, Environmental and Nutritional Sciences (DeFENS), Division of Human Nutrition, Università degli Studi di Milano, 20133 Milan, Italy; marino@unimi.it

**Keywords:** human milk, fatty acids, preterm, growth velocity

## Abstract

This study aimed to analyze the fatty acid content in human milk and to find its relationship with the growth velocity of preterm infants. Mature milk samples from 15 mothers of preterm infants were collected from three different hospitals, followed by lipid extraction, fatty acid methylation, and finally gas chromatography analysis to determine the fatty acids composition. The average total lipid content was 3.61 ± 1.57 g/100 mL with the following classes of fatty acids: saturated fatty acids 43.54 ± 11.16%, unsaturated fatty acids 52.22 ± 10.89%, in which monounsaturated fatty acids were 36.52 ± 13.90%, and polyunsaturated fatty acids were 15.70 ± 7.10%. Polyunsaturated fatty acid sub-class n-6 was 15.23 ± 8.23% and n-3 was 0.46 ± 0.18%. Oleic acid, palmitic acid, and linoleic acid were the most abundant fatty acids. The n-6/n-3 ratio was 32.83:1. EPA and DHA fatty acids were not detected. As gestational age and birth weight increase, C20:2n6 content increases. The growth velocity increases with the decrement in C16 and increment in C20:2n6. The lipid profile of preterm human milk was found to be low in some essential fatty acids, which may affect the quality of preterm infants’ nutrition.

## 1. Introduction

Breastfeeding is the most common and perfect way in which newborn infants can receive the needed nutrients that help in achieving healthy growth and development in the early stages of life. It is recommended to exclusively breastfed infants for the first hour after delivery and the first 6 months of their life, and then it is advised to continue breastfeeding for 2 years along with the introduction of food [1].

The major source of energy in breastfed infants is fat, which is present in the form of lipid globules that contribute to 40–55% of the total intake of energy [2]. Triacylglycerol (TAG) is the major lipid form present in human milk comprising 98–99% of total lipids. Each TAG consists of a glycerol molecule that is bonded to three fatty acid chains of different lengths. These fatty acids (FAs) can either be saturated fatty acids (SFA) that are characterized by having no double bond in their hydrocarbon chain, monounsaturated (MUFA) that have a single double bond, or polyunsaturated (PUFA) that have more than one double bond in their chain [3]. There are three possible sources of FA in breast milk, and they can come from the maternal diet or mothers’ adipose tissues or can be synthesized in mothers’ body tissues [4].

Long-chain polyunsaturated fatty acids (LC-PUFAs), which have ≥18 carbons in the chain, play important roles during the pregnancy of the woman and after the birth for both mother and infant [5]. During pregnancy, LC-PUFAs ensure the proper growth of the fetus and reduce the chances of preterm birth [6], whereas after birth they help in regulating growth, developing the immune system functions, improving the allergic and inflammatory responses, and developing the motor and the nervous system [7,8].

As shown by different references, infants born smaller than average are more likely to face certain diseases during their life [1]. Preterm infants, who are born less than 32 weeks of gestation, require higher amounts of nutrients than term infants due to the immaturity of their gastrointestinal tract and rapid growth rate [9,10]. Mother’s milk has proven to be the most suitable feeding strategy for preterm infants [11]. It was found that human milk reduced the incidence of necrotizing enterocolitis (NEC) and sepsis during their hospital stay [12]. Furthermore, it reduced the morbidity and mortality rates at NICUs. In addition, it enhanced the neurodevelopment of these infants [13]. Due to its variability between mothers and within different lactation periods, infants may suffer from a shortage in their nutrition [14]. Dror and Allen (2018) suggested that human milk lipid content may not change with the maternal diet, but fatty acids can be modified [15]. Innis (2004) found that unsaturated FAs are highly affected by the quality of maternal intake of FAs [16]. Little is known about the effect of different FAs on preterm infants’ growth, and such knowledge is needed to improve the feeding and fortification strategies used in neonatal intensive care units (NICUs) to achieve the best growth outcomes. This study aims to assess the FA profile of human breast milk from mothers who delivered preterm infants and to assess its effect on preterm infant growth velocity for 30 days of their hospital stay. To our knowledge, this is the first study in the Kingdom of Bahrain that addresses this issue and analyzes human milk for its fatty acids. This will provide baseline data for future research.

## 2. Materials and Methods

### 2.1. Study Design

This study is a continuation of the pilot study of Ahmed et al. (2021), which aimed to assess the effect of human milk energy and macronutrient content on preterm infants’ growth rate [17]. It was conducted between July 2018 and March 2021 at the NICUs of the three main hospitals in the Kingdom of Bahrain, Salmaniya Medical Complex (SMC), Bahrain Defence Force Royal Medical Services (BDF) and King Hamad University Hospital (KHUH). Fifteen lactating mothers of healthy preterm infants, born with a birth weight less than 1500 g or with a gestational age of less than 32 weeks, and receiving their mother’s milk, participated in this study. Basic data of infants, including gestational age (weeks), birth weight (kg), daily weight (kg), and weekly head circumference (cm), were collected from their medical records. Growth velocity (GV) for 30 days or until discharge was calculated in g/kg/day as suggested by Patel et al. (2009) [18], and head circumference growth rate (cm/week) was calculated by taking the average of the readings for 4 weeks or until discharge. All this was started when infants reached full feed of 120 mL/kg/day of expressed breast milk and discontinuation of intravenous infusion. Some infants received fortified expressed breast milk and preterm formula milk as prescribed by their physician for a few days during the study.

### 2.2. Sample Collection

Milk samples were collected two weeks after delivery by the mothers when the milk composition was more stable (mature milk) [19], either at home or at the hospital using electrical pumps. The samples were collected once a week for only two constitutive weeks. Two samples, 5 mL each, were collected each day (from morning and evening expressions) and pooled in one breast-milk storage bag to exclude diurnal variation in milk composition. Milk samples were stored at −100 °C freezer until analysis. The samples were thawed at 37 °C water bath and vortexed for 30 s before analysis.

### 2.3. Lipid Extraction and Methylation

Total lipids were first extracted from the milk using the Folch et al. (1957) method [20]. Nineteen ml of (2:1) chloroform–methanol solution was added to 4 mL of the milk sample and vortexed for 10 min, and then 4 mL of deionized water was added, and the tubes were vortexed again for 10 min. To separate the mixture into two phases, centrifugation was performed at 2400 rpm for 20 min. In a pre-weighed tube, the lower phase was transferred and dried under the nitrogen gas in a 37 °C water bath until a clear oil was formed. Total lipids were expressed as g/100 mL of human milk.

This was followed by FA methylation according to the modifications that were conducted by Ozogul et al. (2012) to the Association of Official Analytical Chemists (AOAC) procedures (1990) [21,22]. The whole extracted milk lipid was mixed with 1.5 mL of 0.5 M methanolic sodium hydroxide, and then heated for 7 min at 100 °C and left to cool down at room temperature. After the addition of 2 mL of 14% boron trifluoride-methanol to the mixture, it was heated for 5 min at 100 °C and cooled down to 30–40 °C. One ml of isooctane was added to the tube and vortexed for 30 s. Immediately, 5 mL of saturated sodium chloride was added, and shaking was applied for 30 s. The tube was left to allow the separation of the layers. The isooctane upper layer was placed in another tube and 1 mL of isooctane was added to the mixture again and vortexed for 30 s, and it was left to separate, and the upper layer was removed. After combining the two isooctane extract layers, they were dried under nitrogen gas to 1 mL.

### 2.4. Gas Chromatography

A Perkin Elmer, Clarus 500 GC-FID gas chromatography (GC) equipped with Flame Ionization Detector (FID) and Thermal Conductivity Detector (TCD) was used for the analysis of fatty acids. Using a 30 m × 0.25 mm × 0.25 µm fused carbon-silica column with a temperature range from 40 °C to 260 °C (Stabilwax, Crossbond, Carbowax, Polyethylene glycol), the individual fatty acid methyl esters (FAMEs) were separated. The temperatures were set at 200 °C for the GC oven, 300 °C for FID, 150 °C for TCD, and 250 °C for the injector with a split ratio of 1:2. The flow rate for the carrier gas (nitrogen) was 0.76 mL/min and for the other gases was 450 mL/min for air and 45 mL/min for hydrogen. The injection volume of the sample was 5 µL and the sampling rate was 12.5 Hz. Eighty minutes was set as the total run time for each sample. PUFA No.1 (Marine source) and PUFA No.2 (Animal source) supplied by SUPELCO (USA) were used as an authentic FAMEs standard for identification according to the retention time. FAs to be detected were like the PUFAs used, which ranged from C14 to C24. Values of FAs were expressed as mg/100 mL of human milk and g/100 g of FAs (%).

### 2.5. Statistical Analysis

The data were processed in Microsoft Excel 2010 (Microsoft Corp., Redmond, WA, USA). The normal distribution of the variables was checked using the Shapiro-Wilks test and using Q-Q graphs.

Results of basic characteristics and FA classes were reported as mean ± standard deviation (SD), while FA profiles were reported as mean, variance, SD, minimum (Min), and maximum (Max) in mg/100 mL of human milk and % of fatty acids. Jeffrey’s Amazing Statistics Program (JASP) (JASP Team (2022). JASP (Version 0.16.2) [Computer software]) was used to find the effect of gestational age, birth weight, and GV, on the FA composition of human milk using Kendall’s tau-b correlation coefficient (r). Scattered plots were used to demonstrate the significant correlations. *p*-values of <0.05 were set as significance level.

## 3. Results

### 3.1. Descriptive Data

The basic characteristics of 19 preterm infants of the participated mothers and the mean total lipid content of collected milk samples are described in Table 1. Most of the infants were males (n = 11) and twins (n = 11). Fourteen of them were delivered by cesarean section and the rest were delivered by normal or spontaneous vaginal delivery. The GV for only 4 preterm infants was calculated for 15 days instead of 30 days due to discharge. All infants received expressed own mother’s milk every day during the study. Nine infants did not receive any fortified milk or formula milk. The average number of days infants received fortified milk was 6.3 days, while the average number of days infants received formula milk was 8.6 days. More demographic and anthropometric data of the recruited preterm infants are present in the study of Ahmed et al. (2021) [17].

### 3.2. Classes of Fatty Acids in Human Milk

Table 2 summarizes the sum of SFAs, unsaturated fatty acids (UFAs), MUFAs, PUFAs, omega-6 (ω-6, n-6), and omega-3 (ω-3, n-3) that were found in human milk in two units (mg/100 mL) and (%).

The data show that the UFA class is greater than the SFAs, accounting for more than half the total lipids. In the UFAs, the MUFAs were the highest, accounting for 69.94% of the UFAs, followed were the PUFAs with 30.07% of UFAs. The PUFAs were found to be mostly n-6 FAs (97.01% of PUFAs), while n-3 represented only 2.99% of the PUFAs.

### 3.3. Fatty Acid Profile of Human Milk Samples

Thirteen FAs were found in approximately all milk samples. In the SFAs, (C14:0, C16:0, C20:0, and C18:0) were seen. It was found that the palmitic acid (C16:0) comprised the highest value and (C18:3n6) was the lowest one. In the MUFAs, five FAs were found: (C16:1n9, C16:1n7, C18:1n9, C18:1n7, and C20:1n9), with oleic acid (C18:1n9) being the highest and palmitoleic acid (C16:1n7) being the lowest. In the PUFAs, four FAs were found: (C18:2n6, C18:3n-6, C18:3n-3, and C20:2n6) with linoleic acid (C18:2n6) being the highest value and γ-linolenic acid (C18:3n6) being the lowest. Three n-6 FAs were detected, (C18:2n6, C18:3n-6, and C20:2n6), with the linoleic acid (C18:2n6) being the highest one and γ-linolenic acid (C18:3n6) being the lowest one. Only one n-3 FA was detected, while α-linolenic acid (C18:3n3), Eicosapentaenoic acid (EPA) and Docosahexaenoic acid (DHA) were not detected. The absolute values (mg/100 mL) were converted to relative values (%), where oleic acid (C18:1n9) was the highest and γ-linolenic acid (C18:3n6) was the lowest (Table 3).

Figure 1 represents the significant correlations found between birth weight, gestational age at birth, and growth velocity of preterm infants and the FA content of the milk samples. Positive correlations were found between birth weight and gestational age with eicosadienoic acid (C20:2n6) (r = 0.404, *p* = 0.017; r = 0.563, *p* = 0.001), respectively. A strong negative correlation was found between growth velocity and C16:0 FA (r = −0.538, *p* = 0.001), but positively correlated to C20:2n6 (r = 0.491, *p* = 0.004). No other significant correlations were found between birth weight, gestational age at birth, and GV with the rest of the FAs.

## 4. Discussion

The lipid profile of preterm human milk was found to be low in some essential FAs, which may affect the quality of preterm infants’ nutrition.

The SFAs detected in this study were the medium-chain FAs (C14:0), long-chain FAs (C16:0, C18:0), and very long-chain FAs (C20:0). In the present study, the sum of SFAs (43.54 ± 11.16%) was close to the study conducted by Thakkar et al. (2019) (45.88 ± 7.45%) [23]. By contrast, Wan et al. (2010) and Miliku et al. (2019) found a lower value (35.92 ± 7.34%, 39.75 ± 5.00%, respectively) [24,25].

Palmitic acid (C16:0) was the dominant SFA in the present study (26.12 ± 3.37%), which was consistent with the above studies. A negative correlation was found between palmitic acid (C16:0) and growth velocity. Palmitic acid can be endogenously synthesized from glucose in the liver [26]; because the gestational age increases, the development of the body organs will be fully completed and they will function accurately, and hence the functional requirement for the palmitic acid will be decreased. It was found that 10–12% of the total energy intake comes from palmitic acid [27]. The study by Ahmed et al. (2021) found that preterm infants that were fed human milk with less total lipids than the recommended amount (4.4–6.0 g/100 kcal) had better weight gain rates, most probably due to the reduced protein oxidation by lipids as a result of using carbohydrates as an energy source instead of lipids [17,28].

The MUFAs accounted for (36.52 ± 13.90%) of the total FA content, which was slightly higher than the study conducted by Wan et al. (2010) with a value of 32.59% (±7.21%) [24]. However, it was lower than the studies by Miliku et al. (2019) and Thakkar et al. (2019) with the values of 43.06% (±3.59%) and 40.44% (±5.6%) [23,25]. Overall, the MUFAs were the most abundant FAs in the UFA class in all these studies including the present study. Oleic acid (C18:1n9) was found to be the most abundant MUFA (32.12 ± 4.08%), which was similar to the studies by Wan et al. (2010) (31.26 ± 3.72%), Miliku et al. (2019) (37.05 ± 3.59%), and Thakkar et al. (2019) (35.22 ± 5.16%), comprising 88% of the total MUFAs [23,24,25]. The PUFAs comprised 15.70% (±7.10%) of the total FAs, which is similar to the value found by Koletzko (2016) (15.2 ± 4.26 %) Bzikowska-Jura et al. (2019) (15.1 ± 3.4%) and Freitas et al. (2020) (14.94 ± 5.07%) [29,30,31]. A positive relationship was found between the gestational age (r = 0.563, *p* ˂ 0.001), birth weight (r = 0.404, *p* < 0.017), and growth velocity (r = 0.491, *p* ˂ 0.004) with Eicosadienoic acid (C20:2n6 %). It was found that n-6 LC-PUFA had a protective effect against intestinal injury in the murine model, reducing inflammation and intestinal damage by increasing the lipoxin A4 levels [32]. This may explain the need for this FA for preterm infants who have immature intestines and help to reduce the incidence of NEC.

Only one n-3 FA, namely α-linolenic acid (C18:3n3), was found in the current study; this finding was not in agreement with other studies where they found more than one n-3, including EPA (C20:5n3) and DHA(C22:6n3) [25,29]. Very few studies could not detect the EPA and DHA; one of them is the study conducted in Brazil by Freitas et al. (2020) where the only n-3 FA found was α-linolenic acid (C18:3n3), which is consistent with this study [31].

A study conducted by Bzikowska-Jura et al. (2019) investigated the association between the n-3 FA levels and their maternal current dietary intake and habitual dietary intake. They found that the frequency of food products and fish intake had a positive correlation with the concentrations of α-linolenic acid, EPA, and DHA in human milk [30].

Miliku et al. (2019) found that milk samples from mothers who were not taking fish oil supplements had the lowest EPA and DHA levels (0.07% and 0.16%) in comparison to mothers who were taking them during pregnancy (0.10% and 0.23%) and lactation (0.13% and 0.27%). Their study also showed that the fish oil supplements had a positive association with the high n-3 pattern and negative associations with the high n-6 pattern and n-6/n-3 ratio, where the ratio changed from 6.63% with no supplementation to 6.14% with supplementation (*p* < 0.01), and similar results were found for the intake of shellfish, white fish and fatty cold-water fish [25].

A study conducted in the Kingdom of Bahrain by Freije (2009) showed that the EPA and DHA levels in local fish ranged from 0.030 to 0.239 g/3oz, which was much lower than the range recommended by the USDA Nutrient Data Laboratory (0.13–1.81 g/3oz), and this accounts for the low EPA and DHA intake (0.02–0.13 g/day) that does not meet the recommended value (0.3–0.5 g/day) [29]. Accordingly, the absence of EPA and DHA in the present study can be attributed to the low maternal intake frequency of foods high in n-3 (fish and seafood), the low n-3 quality in the local fish, as well as not taking n-3 supplements during pregnancy and the lactation period [25,30,33].

The GV of the study sample was found to be less than the recommended values (GV = 15–20 g/kg/day) [34]. Much et al. (2013) found that 3-n LCPUFA is responsible for increasing fat mass over the first year of a newborn’s life. Unfortunately, it was possible to compare 3-n LCPUFA because EPA and DHA were missing. However, this can also explain the low GV of the study subjects [35].

In the present study, the n-6/n-3 ratio was 32.83:1, which was much higher than the other studies, such as those by Thakkar et al. (2019) (9.72 ± 4.94%) and Miliku et al. (2019) (6.53 ± 1.72%), and this was due to the very low sum of n-3 fatty acids (0.46 ± 0.18%) found in this study in comparison to the previous studies by Thakkar et al. (2019) (1.15 ± 0.62%), Wan et al. (2010) (1.22 ± 0.29%), and Miliku et al. (2019) (2.39 ± 0.7%) [23,24,25].. The high n-6 in the Westernized diet can be due to the evolution of modern agriculture, agribusiness, processed food, and the production of hydrogenated and refined vegetable oil [36,37,38,39,40].

It is important to inspect the role of eicosadienoic acid (C20:2n6) in preterm growth and quantify the best amount needed to obtain the best growth outcome for infants. This information can be used for the improvement of a lactating mother’s diet or the introduction of n-3 supplements to reduce the n-6/n-3 ratio. In addition, it can be used in the manufacturing of milk fortifiers and preterm infants’ formula milk. Additionally, the infants included in this study did not receive any antibiotics or probiotics during the observation.

The main limitation of the study is the low sample size of the subjects due to the low number of eligible mothers and preterm infants for the study criteria and the low participation rate. A study with a larger sample size is recommended to be conducted before the implementation of the recommendations. The essential FAs such as EPA and DHA were missing, which prevents the human milk from being sufficient for providing the optimum nutrition required for the growth and development of the preterm infant. Another limitation is that it was difficult to compare results with other studies that use different units of measurement (ounces instead of grams).

## 5. Conclusions

To our knowledge, this is the first study conducted in the Kingdom of Bahrain to evaluate human milk. The lipid profile of preterm human milk was found to be low in some essential FAs, which may affect the quality of preterm infants’ nutrition. Eicosadienoic acid (C20:2n6) was found to positively affect the growth velocity of preterm infants, which may be due to reducing inflammation and intestinal injury. Another key finding was that a negative correlation was found between palmitic acid (C16:0) and growth velocity which could be due to reduced protein oxidation by lipids. Unfortunately, EPA and DHA were not detected, which indicates that preterm infants may receive milk poor in n-3 FA. Mothers need to increase their n-3 FA intake either by food or supplements to eliminate its deficiency. Further investigation with a larger sample size is needed to confirm the study’s findings.

## Figures and Tables

**Figure 1 children-10-00939-f001:**
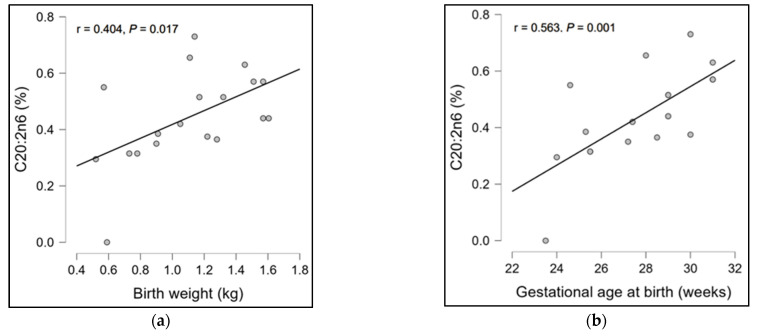
The effect of (**a**) birth weight, (**b**) gestational age at birth, and (**c**,**d**) growth velocity on the fatty acid content of mother’s milk in g/100 g (%) of total fatty acids.

**Table 1 children-10-00939-t001:** Preterm infant’s basic characteristics.

Characteristic	Mean ± SD
Gestational age at birth (weeks)	28.09 ± 2.33
Birth weight (kg)	1.14 ± 0.353
Growth velocity (GV) (g/kg/day)	13.85 ± 4.10
Head circumference growth rate (HC) (cm/week)	0.63 ± 0.18
Total lipids (g/100 mL)	3.61 ± 1.57

SD: Standard deviation.

**Table 2 children-10-00939-t002:** The sum of different classes of fatty acids in human milk.

Fatty Acids Classes	Mean ± SD mg/100 mL	Mean ± SD %
Σ SFAs	852.10 ± 234.80	43.54 ± 11.16
Σ UFAs	1079.02 ± 227.05	52.22 ±10.89
Σ MUFAs	761.02 ± 291.28	36.52 ± 13.90
Σ PUFAs	318.00 ± 142.92	15.70 ± 7.10
Σ n-6	307.98 ± 165.59	15.23 ±8.23
Σ n-3	10.02 ± 6.84	0.46 ± 0.18
n-6/n-3 ratio	30.73:1	32.83:1

MUFAs: Monounsaturated fatty acids, n-3: Omega-3, n-6: Omega-6, PUFAs: Polyunsaturated fatty acids, SD: Standard deviation, SFAs: Saturated fatty acids, UFAs: Unsaturated fatty acids.

**Table 3 children-10-00939-t003:** The fatty acid profile in the milk samples of the preterm infant that was collected postpartum expressed as (mg/100 mL) of milk and % of fatty acids.

Fatty Acids	mg/100 mL of Human Milk					% of Fatty Acids				
	Mean	Var	SD	Min	Max	Mean	Var	SD	Min	Max
14:0	211.60	6848.97	82.76	61.96	335.51	11.67	17.70	4.21	6.03	19.01
16:0	526.41	54,281.49	232.98	114.12	955.30	26.12	11.37	3.37	19.82	31.47
16:1n9	50.83	769.71	27.74	10.31	103.08	2.51	0.55	0.74	0.57	3.41
16:1n7	6.04	111.35	10.55	0.78	44.97	0.27	0.18	0.42	0.09	1.85
18:0	107.91	2380.59	48.79	27.36	179.40	5.43	0.79	0.89	3.93	7.06
18:1n9	672.33	113,088.66	336.29	103.54	1301.31	32.12	16.62	4.08	26.04	41.30
18:1n7	21.45	221.37	14.88	6.43	66.54	1.13	0.24	0.49	0.38	1.93
18:2n6	293.85	20,688.57	143.84	42.66	550.22	14.58	10.38	3.22	11.01	21.42
18:3n6	5.17	22.55	4.75	0.00	20.35	0.22	0.03	0.18	0.00	0.84
18:3n3	10.02	46.82	6.84	1.32	28.36	0.46	0.03	0.18	0.27	0.90
20:0	6.18	10.68	3.27	1.63	13.45	0.33	0.01	0.11	0.18	0.61
20:1n9	10.37	29.50	5.43	0.00	22.36	0.50	0.04	0.20	0.00	0.82
20:2n6	8.96	22.40	4.73	0.00	18.96	0.43	0.03	0.18	0.00	0.73

Max: Maximum value, Min: Minimum value, SD: Standard deviation, Var: Variance, Mean: mean of cases.

## Data Availability

Not applicable.

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
