# Peer review of "Human Milk Fatty Acid Composition and Its Effect on Preterm Infants’ Growth Velocity"

_children, 2023, doi:10.3390/children10060939_

Round 1

Reviewer 1 Report

First of all, I want to congratulate the authors for this study.

I would like to know if the preterm infants included in your study were delivered by caesarean section or vaginal delivery? Can you specify if the preterm infants received antibiotics or probiotics?

Author Response

QUESTION: I would like to know if the preterm infants included in your study were delivered by caesarean section or vaginal delivery? Can you specify if the preterm infants received antibiotics or probiotics?

ANSWER: Thanks a lot for your valuable comment. Following your input, we have mentioned that 14 preterm infants out of 19 were delivered by caesarean section. Only 5 were delivered by normal or spontaneous vaginal delivery. In according to the medical records and the specialists, the infants did not receive any antibiotics or probiotics.  We mentioned this statement into the text following your valuable suggestion.

Reviewer 2 Report

The article entitles “Human Milk Fatty Acid Composition and its effect on Growth Velocity of Preterm Infants” (children-2398557) has been submitted to the "Pediatric Neonatology" section of the Special Issue "10th Anniversary of Children: Feature Papers in Neonatology" in the journal "Children".

 The purpose of this article is to examine the fatty acid composition of human milk and its relationship with the growth velocity of preterm infants. However, some issues need to be addressed. Firstly, the title of the paper suggests a relationship between the fatty acid composition of human milk and the growth velocity of preterm infants, but the study only examines the milk composition of mothers who gave birth to preterm infants up to 30 days or until discharge. The follow-up period needs to be clarified in the paper. Additionally, the sample size is small, and it is unclear if the statistical analysis took into account the normality of the distribution of the quantitative variable.

The study describes the quality of milk in relation to fatty acids, but we would expect a discussion on the relationship between the fatty acid composition of human milk and the growth of premature infants. Furthermore, Figure 1 presents correlations with a wide dispersion, and each correlation is associated with a different type of fatty acid. The authors should provide an explanation for this situation.

The discussion should consider the study's limitations, such as the difficulty in comparing results with other studies that use different units of measurement (ounces instead of grams). The conclusions should be adjusted to the results obtained.

Overall, the article needs to be revised to address these issues and to provide a clear and accurate representation of the study's findings.

Author Response

QUESTION The follow-up period needs to be clarified in the paper. Additionally, the sample size is small, and it is unclear if the statistical analysis took into account the normality of the distribution of the quantitative variable.

ANSWER: Thanks a lot for your valuable suggestions. Each comment was valuable and improved a lot our paper.

 Following your comments (a) we included in the statistical analysis section the normality calculation. (b) the small sample size  has been take in account into the discussion as limitation, (c) the follow up period has been clarified in an ad hoc paragraph.

QUESTION The study describes the quality of milk in relation to fatty acids, but we would expect a discussion on the relationship between the fatty acid composition of human milk and the growth of premature infants. Furthermore, Figure 1 presents correlations with a wide dispersion, and each correlation is associated with a different type of fatty acid. The authors should provide an explanation for this situation.

ANSWER: The correlation between all fatty acids was inspected with the other parameters, but only significant relationships were represented in Figure 1 to reduce the space that will be taken if we represented all the figures, and it is much easier to see the significant results.

QUESTION The discussion should consider the study's limitations, such as the difficulty in comparing results with other studies that use different units of measurement (ounces instead of grams). The conclusions should be adjusted to the results obtained.

ANSWER: Study limitations have been included extensively in the discussion. Also we included the suggested statement regarding the unit of measure.

Reviewer 3 Report

This paper is well-written and has clinical importance. The authors found that growth and birth weight of preterm infants correlated with the C20:2n-6 content, and inversely correlated with the C16:0 content. I really want to see this paper accepted however I have minor criticism:

In Table 3 authors need to add an additional column that describes the functions of listed fatty acids.

Author Response

QUESTION In Table 3 authors need to add an additional column that describes the functions of listed fatty acids.

ANSWER: Thanks a lot for your valuable comment. For a better understanding, some of the functions of fatty acids have been mentioned in the introduction and discussion. We preferred avoiding to List them in the table because it will make the manuscript too long.

Round 2

Reviewer 2 Report

I have reviewed the submitted article entitled "Human Milk Fatty Acid Composition and its effect on Growth Velocity of Preterm Infants" (children-2398557), as well as the reviewers' response.

I believe that the article has improved in terms of communicating the information after incorporating the suggested changes.

However, I think that in Table 3, the last two rows should be removed as they do not provide any informative content.